# Identifying Core Items of the Japanese Version of the Mother-to-Infant Bonding Scale for Diagnosing Postpartum Bonding Disorder

**DOI:** 10.3390/healthcare11121740

**Published:** 2023-06-13

**Authors:** Kaori Baba, Yaeko Kataoka, Toshinori Kitamura

**Affiliations:** 1Research Center for Social Science & Medicine, Tokyo Metropolitan Institute of Medical Science, Tokyo 156-8506, Japan; 2Graduate School of Nursing Science, St. Luke’s International University, Tokyo 104-0044, Japan; 3Kitamura Institute of Mental Health Tokyo, Tokyo 151-0063, Japan; 4Kitamura KOKORO Clinic Mental Health, Tokyo 151-0063, Japan; 5T. and F. Kitamura Foundation for Studies and Skill Advancement in Mental Health, Tokyo 151-0063, Japan; 6Department of Psychiatry, Graduate School of Medicine, Nagoya University, Nagoya 466-8550, Japan

**Keywords:** postpartum period, parents, questionnaire, bonding disorder, mother-to-infant bonding scale, longitudinal study, factor analysis, psychometrics

## Abstract

The Japanese version of the mother-to-infant bonding scale (MIBS-J), a self-report of postpartum bonding disorder, is used in clinical settings for screening postpartum bonding disorder at various time points. However, its psychometric properties, particularly measurement invariance, have rarely been reported, and the validity of comparing scores across time points and sex is unclear. We aimed to select and validate the MIBS-J items suitable for parents at three time points. Postpartum mothers (n = 543) and fathers (n = 350) were surveyed at 5 days, 1 month, and 4 months postpartum. All participants were randomly divided into two subgroups, one for exploratory factor analyses (EFAs) and another for confirmatory factor analyses. Measurement invariance of the best model was tested using the entire sample, between fathers and mothers, and across the three observation periods. A three-item structure (items 1, 6, and 8) extracted through EFAs showed acceptable configural invariance. This model was accepted for scalar invariance between fathers and mothers and for metric invariance across the three time points. Our findings suggest that the three-item MIBS-J is sufficient for diagnosis of postpartum parental bonding disorder through continuous observation for at least 4 postpartum months, in order to detect the priority of parents who need support.

## 1. Introduction

The emotional tie that a parent feels toward their infant is termed “bonding.” Some mothers experience dislike, resentment, or hatred, and/or desire for permanent abandonment toward their infant and hope for the child to disappear [1,2]; this is called bonding disorder [3]. Postpartum bonding disorders toward infants may have serious and long-term consequences on the child’s development and mother–child relationship [3], such as abusive parenting [4,5,6] and psychiatric or learning disorders in the children [7]. An international position paper on mother–infant (perinatal) mental health [8] notes that “a small minority of mothers may suffer from emotional rejection of the infant, which, together with psychosis and suicidal depression, is in the first rank of severity in this area of psychiatry.” These disorders are more common, intractable, and serious in their effects than puerperal psychosis, but they can be resolved completely with treatment [7]. Hence, it is important to detect the presence and extent of bonding disorders as early as possible in the perinatal period, and several measurements have been developed [9].

Some self-report instruments measure parental bonding disorders, including the mother-to-infant bonding scale (MIBS) [10]. A Japanese modification of the MIBS (MIBS-J), developed by Yoshida et al. [11], has been widely and repeatedly used at different time periods (e.g., during postnatal hospitalization, 1-month check-ups, and 4-month infant check-ups) in clinical and research perinatal settings in Japan (e.g., Ohashi et al. [4] and Baba et al. [5]). The repeated use of a psychological measure is usually accompanied by reduced validity [12,13,14], and the appropriateness of using all the items of the MIBS-J at multiple time points is unclear. Besides, some of the 10 MIBS-J items are quite highly skewed [11,15]. Consequently, most of the general population of parents would report “no problem” to these items. Indeed, there may be little clinical significance in using all 10 items for the general population, as is currently the case. Previous studies have constructed subscales of 7 items [11,15] or 8 items [16].

The MIBS-J factor structure has been reported in at least three studies [11,15,16], but the factor structure was not identified individually for each population (e.g., sex and time period) using both exploratory and confirmatory factor analyses (EFAs and CFAs, respectively) in two of these studies [11,15]. Yoshida et al. [11] conducted a longitudinal study of postpartum mothers (n = 554) and collected responses at three different time points: 5 days, 1 month, and 4 months after childbirth. They further conducted EFAs and CFAs and a two-factor structure (lack of affection [LA] and anger and rejection [AR]) with 7 of 10 items was extracted [11]. However, this procedure could not identify the differences in terms of the factor structure of the scale across the time periods because the data obtained from the three time points were combined to create a single data set. The means of the two subscale scores decreased with time; however, Yoshida et al. [11] did not measure the factor mean invariance across the time points. A few years later, Kitamura et al. [15] collected data using the MIBS-J from a cross-sectional sample of fathers (n = 396) and mothers (n = 733) with children aged 0–10 years. They identified a two-factor structure of the MIBS-J items, which was quite similar to that reported by Yoshida et al. [11]. They found a good fit of the model with the data regarding configural invariance (CFI = 0.956 and root mean square error of approximation [RMSEA] = 0.033). However, they did not report on measurement invariance among mothers and fathers. Motegi et al. [16] collected MIBS-J data from a longitudinal sample of early pregnancy, late pregnancy, and postpartum groups and performed analyses of multiple-group measurement invariance to determine the extent of the factor structure across the different groups, such as early pregnancy and late pregnancy. They identified two-factor structures [16], but the included items were different from previous studies [11,15].

The use of a psychological measure requires confirmation of configural, measurement, and structural invariance of the factor structure of the measure [17,18,19]. Thus, the factor structure of the measure should be stable between participants with different demographic features (e.g., fathers and mothers) and across various observation time points [17,18]; studies adopting these assumptions have increased recently [16,20]. In fact, bonding disorders are a problem not only for mothers, and research on bonding should also focus on fathers [21,22,23,24,25,26,27]. Hence, the target population should be expanded to include fathers, and the measurement invariance of using the MIBS-J among perinatal fathers should be investigated. If the measurement invariance does not hold, items of the measure do not have the same meaning and may cause bias in the assessment, and a comparison of the measured scores does not make any sense. Validating the measurement invariance of the MIBS-J for perinatal mothers and fathers would allow clinicians and researchers to compare the degree of perinatal bonding between parents. The stability of the factor structure is confirmed through several steps [28], and the subsequent steps are endorsed only when the preceding steps are accepted. If one step is rejected, the next step should not be performed. Basic stability is termed configural invariance. Each group (e.g., fathers vs. mothers) should have the same pattern of items and factors (first step). Moreover, factor loadings for similar items (metric invariance, also known as weak factorial invariance; second step), intercepts of similar items (scalar invariance, also known as strong factorial invariance; third step), residuals (errors) of similar items (residual invariance, also known as strict factorial invariance; fourth step), variances of similar factors (factor variance invariance; fifth step), and the means of factors (factor mean invariance; sixth step) should be invariant across groups. The second to fourth steps are termed measurement invariance, and the fifth and sixth steps are termed structural invariance.

Understanding parents’ bonding disorders is an important aspect in the field of perinatal psychiatry [8], and focusing on stable scale items that are independent of each time period and sex is essential. Therefore, the aim of this study was to extract the MIBS-J items appropriate for mothers and fathers at three time points (postpartum hospitalization period,1 month postpartum, and 4 months postpartum) commonly used in Japanese clinical practice in order to continuously assess bonding disorders among perinatal mothers and fathers over time.

## 2. Materials and Methods

### 2.1. Procedures and Participants

We recruited mothers and fathers on days 3–5 postpartum at the maternity ward of one perinatal medical center, three general hospitals, two antenatal clinics, and one birth center in Tokyo and its suburban areas. The inclusion criteria were as follows: (a) a good command of the Japanese language, (b) residing in Japan, (c) no serious physical diseases or pregnancy-related complications, and (d) a singleton neonate. One of the investigators (KB) visited the wards and recruited participants after explaining the study and obtaining written informed consent from the participants. Data were collected in three waves: 5 days (Wave 1, W1), 1 month (Wave 2, W2), and 4 months (Wave 3, W3) after childbirth. The numbers of fathers and mothers who returned the questionnaire were 421 and 684 at W1, 361 and 590 at W2, and 351 and 566 at W3, respectively. We handed (W1) or posted (W2 and W3) the set of questionnaires to the participants and asked them to return them via the postal service. Mothers and fathers were asked to complete the questionnaire independently. Data were collected from December 2015 to June 2016 as part of the first author’s PhD dissertation. [29] The mean ± standard deviation age of the mothers and fathers was 33.1 ± 4.7 and 34.6 ± 5.1 years, respectively. More than three-quarters (n = 437) of the mothers underwent vaginal delivery, and 18% (n = 98) underwent Cesarean delivery. There were 288 (53%) primiparas and 253 (46.6%) multiparas.

### 2.2. Measurement

#### Mother-to-Infant Bonding Disorder

The MIBS-J comprises 10 items that assess mothers’ attitudes and emotions toward their infants [11]. These items are rated on a 4-point Likert scale (0–3), and higher scores indicate that the mother has a more negative attitude and emotion toward the infant. Two subscales were proposed by Yoshida et al. [11] and Kitamura et al. [15]: LA and AR. LA items include “feel protective toward my baby” (reverse item) and “feel close to my baby” (reverse item), whereas AR items include “feel angry with my baby” and “feel resentful toward my baby”. We used the same questionnaire for the fathers.

### 2.3. Data Analysis

The participants who returned the questionnaire at all three time points (350 fathers and 543 mothers) were randomly split into two subgroups. The first (166 fathers and 282 mothers) and second (184 and 261 mothers) groups were used for EFAs and CFAs, respectively. Missing MIBS-J data were considered to be missing completely at random (Little’s missing completely at random test for fathers [*p* = 0.582] and mothers [*p* = 0.229]) [30]. The missing data were handled by pairwise deletion for all analyses, except for CFAs. For CFAs, the missing data were handled using the full information maximum likelihood method.

In an EFA using the first half sample, we calculated the skewness and kurtosis of all the MIBS-J items. When excessive skewness or kurtosis was present, the MIBS-J items were log-transformed [31]. Items that showed excessive skewness (>4.0) or kurtosis (>15.0), even after log-transformation, were excluded from further analyses [31]. We performed EFAs for the remaining items after conforming to the Kaiser–Meyer–Olkin index of sampling adequacy [32] and Bartlett’s test of sphericity [33] to examine the adequacy of the sample size and non-zero correlations between items [34]. The number of factors was determined using a scree plot. The minimum acceptable factor loading was 0.30 [35], maximum likelihood extraction was performed, and the axes were rotated using Promax rotation.

Next, using the second half of the sample, CFAs were performed to obtain the adequate MIBS-J model extracted by the EFAs. Measures of goodness-of-fit included chi-square (CMIN), CFI, and RMSEA. A good fit was defined as CMIN/degree of freedom (*df*) < 2, CFI > 0.97, and RMSEA < 0.05 [36], and an acceptable fit was defined as CMIN/*df* < 3, CFI > 0.95, and RMSEA < 0.08 [36,37]. However, when the sample size is relatively small (<500) or a model is complex, these criteria might be stringent, and the use of more flexible criteria is suggested (e.g., CFI > 0.90 and RMSEA < 0.10) [38]. We also considered these flexible criteria when examining model fit.

Measurement invariance of the best model was tested with the full sample (participants who returned the questionnaire at all three time points; 350 fathers and 543 mothers), between fathers and mothers, and across the three observation periods. A series of hierarchical models was tested as follows. First, configural invariance was tested. Once configural invariance was supported, which indicates that both sexes or time periods share the same factor structure, metric invariance was tested. When metric invariance was held, which indicates that the factor loadings were equivalent across sexes or periods, scalar invariance was assessed by restricting the item intercepts to be equal across sexes or periods. When scalar invariance was supported, a comparison of residual invariance between sexes and time periods was implemented to examine residual invariance. In addition, structural invariance was needed as evidence of factor structure robustness, and it included factor variance invariance. If one of the above steps was rejected, subsequent steps were not performed. Invariance from one step to the next was “accepted” if we noticed either: (a) a non-significant increase in χ^2^ for *df* of difference, (b) CFI < 0.01, or (c) RMSEA < 0.01 [39,40]. The CFI and RMSEA may be better indicators of measurement invariance than χ^2^ because χ^2^ is sensitive to the sample size and may thus produce excessive “rejection” rates. In addition, since scalar-level measurement invariance is rarely confirmed [41] and there is no consensus on the level to which it should be confirmed, we considered measurement invariance to be present if it is confirmed up to at least metric invariance. To determine measurement invariance, a distinction was made between full and partial invariance [28,42,43].

All statistical analyses were conducted using IBM SPSS 26 and Amos 26 (IBM Corp., Armonk, NY, USA).

### 2.4. Ethical Considerations

This study was approved by the Ethical Committee of St. Luke’s International University (approval no. 15-074).

## 3. Results

### 3.1. Factor Structure Derived from EFA

In the first half group, many MIBS-J items showed high skewness and kurtosis (Table 1). Skewness and kurtosis were less severe after log transformation; however, there were still five items (items 3, 4, 5, 7, and 9) with skewness > 4.0 and kurtosis > 15.0. Hence, we excluded those five items, and the remaining five MIBS-J items were entered into an EFA. The Kaiser–Meyer–Olkin index was 0.74–0.75, and the Bartlett’s test was χ^2^ (*df*) = 171.99 (10)–337.98 (10) (*p* < 0.001). Therefore, the datasets were suitable for EFAs. The EFAs for the fathers and mothers were performed separately. The scree test suggested either a one- or two-factor solution for both fathers and mothers. In the one-factor model, all the MIBS-J items, except item 2, showed factor loadings > 0.3 at all three time points (Appendix A). In the two-factor solution, the first factor was loaded highly (>0.3) on the MIBS-J items 1, 6, 8, and 10, which reflect LA, as demonstrated in previous studies [11,15]. The second factor was loaded highly only on item 2, which reflects AR. We excluded this item from further analyses because a factor with only a single indicator having a high factor loading is unstable for a measurement model.

The remaining four MIBS-J items (items 1, 6, 8, and 10) belonged to the LA category. A single-factor EFA showed high factor loadings for all items at all time points among both fathers and mothers (Table 2).

### 3.2. Measurement Invariance across Sexes or Time Periods

When comparing the factor models between fathers and mothers, configural invariance was accepted. However, metric invariance was rejected for W3 (Appendix A). Therefore, we examined the z value, which indicates the group differences in the factor loadings, and found that item 10 showed the largest group difference (z = 3.165, *p* < 0.01). Therefore, after excluding item 10, we re-examined the EFA and configural invariance (Table 3).

The remaining model had a three-item structure (items 1, 6, and 8). When comparing this model between fathers and mothers, configural invariance was confirmed at all three time points. Measurement invariance also conformed to the stability of factor variance at W1 and W2 and up to scalar invariance at W3 (Table 4).

When comparing the three time points, configural invariance was confirmed in both fathers and mothers (Table 5). Among fathers, metric invariance was rejected, although partial invariance was supported by freeing the restriction of item 6. Subsequently, scalar invariance was accepted. Among mothers, metric invariance was proven. Scalar invariance was rejected in both fathers and mothers (Table 5).

## 4. Discussion

In this study, configural and measurement invariance between mothers and fathers and across three postpartum time points were observed using only three MIBS-J items. This solves the problem of loss of validity due to repeated use observed in previous studies and clinical situations where the MIBS-J was used [12,13,14]. Moreover, to our knowledge, this is the first study in Japan to also include postpartum fathers.

We suggest that 3 of the 10 MIBS-J items are particularly suitable for mothers and fathers at the three time points commonly used in Japanese clinical practice (postpartum hospitalization period and 1 month and 4 months postpartum) to assess perinatal parental bonding disorders continuously over time. Since perinatal bonding is acquired in part through nurturing contact with the child and can fluctuate physiologically [3,44,45,46], our study result of the instability of the scores for parents in the immediate postpartum period is a convincing result. The MIBS-J was originally developed for perinatal mothers, and there was no sufficient verification that it is versatile enough for fathers, but the results suggest that fathers may have a different concept of what constitutes bonding than mothers. Because bonding is a *parental* emotion towards a baby, fathers also have bonding emotions; hence, it is necessary to provide a tool to measure both maternal and paternal bonding emotions. Here, the MIBS-J may not be a perfect tool to be used in mothers and fathers. Future studies should pay attention to the development of such instruments.

Some recent studies have suggested the need for the development of a new scale [16]. However, the development of a new scale would be a large undertaking and not be immediately feasible. While a new scale is being developed, it may be possible to use the three items presented in this study to observe paternal bonding over time, which is gaining attention not only with mothers, but also with fathers [21,22,23,24,25,26,27]. Three of the 10 MIBS-J items may enable stable comparison of changes in bonding, and clinicians can find priority support targets within the limited support time and human resources (e.g., fathers to be prioritized over mothers).

It should be understood that the present results (the three items extracted were a one-factor structure [LA]) do not reflect the MIBS-J assumption that bonding disorders are composed of two sub-concepts [8]. According to a previous study [3], AR tends to occur in severe cases, which may have influenced the lack of severe cases in this study of the general population. Unlike LA items, AR items are likely to be influenced by some situations, e.g., newborn colic [47,48], which might have reduced the invariance of the instrument. Therefore, the results of this study do not overturn the assumption that the MIBS-J consists of 10 items with two sub-concepts. Our findings highlight the importance of emphasizing the affection items of the MIBS-J to measure parental bonding with stability during the first 4 postpartum months.

Regarding convenience in the population approach, repeated use of all 10 items of the MIBS-J more frequently than once every few months may be burdensome for both the responding parents and observing clinicians. In addition, from a statistical perspective, there is the potential for score bias due to repeated measures [12,13,14]. Therefore, in a population approach such as community care in Japan, using the three items of the MIBS-J may be preferred in the period up to 4 months postpartum in order to detect the priority of parents who need support through continuous observation of parental bonding.

The limitations of this study should be noted. First, several MIBS-J items showed excessive skewness even after log-transformation. This finding suggests that the selection of the participants may have been biased, and the study cohort included few poorly bonded participants, especially those with AR. A population of clinical cases might produce different results. While some opinions are in favor of log-transformation [49,50], it is also true that there are opposing opinions [51,52], so the interpretation of this data should be done with caution. Second, according to a systematic review of scales measuring bonding [9], the psychometric evaluation performance of the MIBS-J is low. Although there are scales measuring paternal bonding (e.g., the Korean paternal–fetal attachment scale and paternal postnatal attachment scale), all have poor psychometric properties. It is desirable to develop a scale with better psychometric properties to measure parental bonding. There is a need for new items or scales that can consistently measure long-term bonding from gestational age to 4 months postpartum and beyond in order to allow trajectory studies of long-term bonding disorders. Third, this study included fewer fathers than mothers; thus, replication studies are needed before a conclusion is reached. Fourth, this study did not test the three items of MIBS-J for reliability and validity or item response theory, which would have shown the amount of information the items can provide; hence, future research needs to address this issue. Fifth, the missing value classification for the present data was confirmed to be missing completely at random (MCAR) by Little’s test and therefore we chose pairwise deletion. Although it does not affect the estimation to the population, the multiple imputation method could have avoided the data reduction due to pairwise deletion. Sixth, it should be noted that the data showing measurement and structural invariance of three MIBS-J items across the three time points (Table 5) showed weak factorial invariance. As noted earlier, we need a common instrument to measure both maternal and paternal bonding emotions with a robust measurement invariance.

The full 10-item MIBS-J is routinely used in Japan for the general population of mothers; however, it may be possible to focus on score changes of the three identified items of the MIBS-J at each time period for comparison, without incurring statistical bias or participant burden, in order to determine support needs.

## 5. Conclusions

Our study confirmed that the three-item MIBS-J was psychometrically robust among Japanese fathers and mothers during the 4-month period after childbirth. All these three items belong to the LA subscale [11,15]. Thus, we believe it is especially important to focus on “affection” to measure parental bonding with stability during the first 4 postpartum months using the MIBS-J in a population approach such as community care in Japan. At present, all 10 MIBS-J items are used in clinical and research settings at various time points during the perinatal period. We suggest that three of the MIBS-J items are particularly suitable for parents at the three time points commonly used in Japanese practice (postpartum hospitalization period and 1 month and 4 months postpartum) to assess perinatal parental bonding disorders continuously over time.

## Figures and Tables

**Table 1 healthcare-11-01740-t001:** Paternal and Maternal MIBS-J Means, Skewness, and Kurtosis at W1, W2, and W3.

Items		Time	Parent	N	Mean	SD	Skewness	Kurtosis	Skewness after Log Transformation	Kurtosis after Log Transformation
1	I feel loving toward my child	W1	Fathers	166	1.25	0.56	2.34	5.40	1.83	2.04
			Mothers	281	1.19	0.48	2.97	10.18	2.25	4.04
		W2	Fathers	166	1.30	0.56	1.70	1.94	1.37	0.31
			Mothers	282	1.18	0.45	2.77	8.60	2.18	3.56
		W3	Fathers	166	1.20	0.42	1.72	1.70	1.59	0.69
			Mothers	282	1.11	0.36	4.08	20.39	3.20	9.71
2	I feel scared or panicky when I have to do something for my child	W1	Fathers	166	1.80	0.73	0.53	−0.28	−0.06	−1.32
			Mothers	279	1.76	0.84	0.89	0.01	0.32	−1.29
		W2	Fathers	165	1.64	0.73	0.77	−0.40	0.35	−1.42
			Mothers	282	1.52	0.68	1.09	0.49	0.63	−1.11
		W3	Fathers	166	1.57	0.65	0.85	0.24	0.35	−1.36
			Mothers	282	1.34	0.62	1.97	4.13	1.32	0.47
3	I feel resentful toward my child	W1	Fathers	166	1.07	0.31	6.28	48.75	**4.70**	**23.82**
			Mothers	281	1.12	0.38	4.22	23.10	3.06	9.15
		W2	Fathers	166	1.20	0.52	3.05	10.64	2.26	4.21
			Mothers	282	1.15	0.39	3.07	12.09	2.38	4.55
		W3	Fathers	166	1.13	0.35	2.68	6.63	2.46	4.46
			Mothers	282	1.13	0.49	4.61	22.76	3.63	13.18
4	I feel nothing for my child	W1	Fathers	166	1.05	0.22	4.26	16.32	**4.26**	**16.32**
			Mothers	280	1.04	0.25	7.76	75.31	**5.85**	**37.48**
		W2	Fathers	166	1.05	0.25	5.06	27.82	**4.55**	**20.23**
			Mothers	282	1.05	0.30	8.07	73.18	**6.46**	**45.03**
		W3	Fathers	166	1.02	0.25	11.38	135.35	**10.32**	**111.38**
			Mothers	282	1.03	0.22	10.18	121.72	**7.94**	**69.72**
5	I feel angry with my child	W1	Fathers	166	1.02	0.13	7.30	51.94	**7.30**	**51.94**
			Mothers	281	1.04	0.21	5.87	37.63	**5.33**	**28.12**
		W2	Fathers	166	1.05	0.25	5.06	27.82	**4.55**	**20.23**
			Mothers	282	1.07	0.26	3.26	8.68	3.26	8.68
		W3	Fathers	166	1.11	0.40	4.43	22.79	3.51	12.13
			Mothers	282	1.07	0.31	5.45	37.04	**4.23**	**18.41**
6	I enjoy doing things with my child	W1	Fathers	166	1.52	0.74	1.32	1.05	0.82	−0.80
			Mothers	279	1.65	0.76	0.93	0.19	0.39	−1.28
		W2	Fathers	166	1.72	0.85	0.93	−0.04	0.42	−1.30
			Mothers	282	1.65	0.74	0.94	0.34	0.37	−1.26
		W3	Fathers	165	1.62	0.78	1.08	0.47	0.55	−1.14
			Mothers	282	1.46	0.67	1.36	1.42	0.84	−0.74
7	I wish my child is different	W1	Fathers	166	1.08	0.42	5.60	32.85	**4.88**	**23.79**
			Mothers	279	1.04	0.22	5.51	32.99	**5.02**	**24.67**
		W2	Fathers	166	1.05	0.34	6.85	49.66	**6.14**	**38.14**
			Mothers	282	1.06	0.25	4.36	19.94	**4.02**	**15.02**
		W3	Fathers	166	1.02	0.19	8.68	80.69	**7.93**	**64.87**
			Mothers	282	1.04	0.25	8.68	85.90	**7.13**	**54.36**
8	I feel protective toward my child	W1	Fathers	165	1.20	0.59	3.42	12.12	2.70	6.50
			Mothers	281	1.12	0.40	3.84	16.48	3.15	9.14
		W2	Fathers	166	1.25	0.54	2.37	5.75	1.83	2.06
			Mothers	282	1.15	0.47	3.74	15.92	2.88	7.67
		W3	Fathers	166	1.24	0.57	2.87	9.18	2.06	3.35
			Mothers	282	1.12	0.46	4.80	25.38	3.72	14.03
9	I wish I did not have my child	W1	Fathers	166	1.05	0.29	7.77	70.82	**6.02**	**39.63**
			Mothers	280	1.10	0.43	5.25	30.15	**4.20**	**18.02**
		W2	Fathers	166	1.05	0.30	7.20	62.03	**5.49**	**32.93**
			Mothers	282	1.10	0.38	4.94	28.60	3.84	**14.92**
		W3	Fathers	166	1.06	0.36	7.11	53.94	**6.01**	**37.93**
			Mothers	282	1.09	0.37	5.21	31.46	**4.09**	**17.07**
10	I feel close to my child	W1	Fathers	166	1.42	0.68	1.45	1.19	1.07	−0.42
			Mothers	280	1.32	0.65	2.22	4.72	1.65	1.42
		W2	Fathers	166	1.54	0.78	1.33	0.96	0.86	−0.76
			Mothers	282	1.23	0.59	2.85	8.02	2.29	4.06
		W3	Fathers	166	1.36	0.61	1.69	2.42	1.22	0.02
			Mothers	282	1.12	0.40	3.73	15.99	2.99	8.19

MIBS-J = Japanese modification of the mother-to-infant bonding scale; W1 = 5 days after childbirth; W2 = 1 month after childbirth; W3 = 4 months after childbirth; SD = standard deviation. Skewness after log transformation > 4.0 or kurtosis after log transformation > 15.0 is presented in boldface; n = 166 for fathers and 282 for mothers.

**Table 2 healthcare-11-01740-t002:** Factor Loadings of Four MIBS-J Items for Each Factor within Different Models.

Items		Time	Parent	Model 1 (1-Factor)
1	I feel loving toward my child	W1	Fathers (n = 166)	**0.69**
			Mothers (n = 282)	**0.70**
		W2	Fathers	**0.81**
			Mothers	**0.85**
		W3	Fathers	**0.71**
			Mothers	**0.76**
6	I enjoy doing things with my child	W1	Fathers	**0.75**
			Mothers	**0.57**
		W2	Fathers	**0.59**
			Mothers	**0.54**
		W3	Fathers	**0.61**
			Mothers	**0.53**
8	I feel protective toward my child	W1	Fathers	**0.58**
			Mothers	**0.65**
		W2	Fathers	**0.74**
			Mothers	**0.66**
		W3	Fathers	**0.79**
			Mothers	**0.65**
10	I feel close to my child	W1	Fathers	**0.68**
			Mothers	**0.68**
		W2	Fathers	**0.65**
			Mothers	**0.68**
		W3	Fathers	**0.75**
			Mothers	**0.64**

MIBS-J = Japanese modification of the mother-to-infant bonding scale; W1 = 5 days after childbirth; W2 = 1 month after childbirth; W3 = 4 months after childbirth; n = 166 for fathers and 282 for mothers. Factor loadings > 0.30 are presented in boldface; the upper figure in each cell represents factor loading (or total variance explained) among fathers, whereas the lower figure in each cell represents factor loading (or total variance explained) among mothers. Item scores after log transformation were entered into an exploratory factor analysis.

**Table 3 healthcare-11-01740-t003:** Factor Loadings of Three MIBS-J Items for Each Factor Within Different Models.

Items		Time	Parent	Model 1 (1-Factor)
1	I feel loving toward my child	W1	Fathers (n = 166)	**0.65**
			Mothers (n = 282)	**0.75**
		W2	Fathers	**0.81**
			Mothers	**0.90**
		W3	Fathers	**0.75**
			Mothers	**0.84**
6	I enjoy doing things with my child	W1	Fathers	**0.77**
			Mothers	**0.55**
		W2	Fathers	**0.56**
			Mothers	**0.48**
		W3	Fathers	**0.65**
			Mothers	**0.50**
8	I feel protective toward my child	W1	Fathers	**0.59**
			Mothers	**062**
		W2	Fathers	**0.76**
			Mothers	**0.65**
		W3	Fathers	**0.73**
			Mothers	**0.59**

MIBS-J = Japanese modification of the mother-to-infant bonding scale; W1 = 5 days after childbirth; W2 = 1 month after childbirth; W3 = 4 months after childbirth; n = 166 for fathers and 282 for mothers. Factor loadings > 0.30 are presented in boldface; the upper figure in each cell represents factor loading (or total variance explained) among fathers, whereas the lower figure in each cell represents factor loading (or total variance explained) among mothers. Item scores after log transformation were entered into an exploratory factor analysis.

**Table 4 healthcare-11-01740-t004:** Measurement and Structural Invariance of Three MIBS-J Items between Fathers and Mothers.

Models		χ^2^	*df*	χ^2^/*df*	Δχ^2^ (*df*)	CFI	ΔCFI	RMSEA	ΔRMSEA	Judgement
W1 fathers vs. mothers	Configural	0	0	0	Ref	1.000	Ref		Ref	Accept
	Metric	4.041	2	2.020	4.041(2) NS	0.929	0.011	0.048	0.048	Accept
	Scalar	11.540	5	2.308	7.499(3) NS	0.902	0.027	0.054	0.006	Accept
	Residual	67.547	15	4.503	23.041(4) ***	0.847	0.055	0.089	0.006	Accept
	Factor variance	67.744	16	4.234	0.197(1) NS	0.849	+0.002	0.085	Δ0.004	Accept
W2 fathers vs. mothers	Configural	0	0		Ref	1.000	Ref		Ref	Accept
	Metric	3.286	2	1.643	3.286(2) NS	0.995	0.005	0.038	0.038	Accept
	Scalar	3.715	5	0.743	0.429(3) NS	1.000	+0.005	0.000	Δ0.038	Accept
	Residual	5.731	8	0.716	2.016(3) NS	1.000	0	0.000	0	Accept
	Factor variance	6.022	9	0.669	0.291(1) NS	1.000	0	0.000	0	Accept
W3 fathers vs. mothers	Configural	0	0		Ref	1.000	Ref		Ref	Accept
	Metric Scalar	1.276 14.800	2 5	0.638 2.960	1.276(2) NS 13.524(3) **	1.000 0.964	0 0.036	0.000 0.067	0.000 0.067	Accept Accept
	Residual	67.874	8	8.484	53.074(3) ***	0.783	0.181	0.130	0.063	Reject

MIBS-J = Japanese modification of the mother-to-infant bonding scale; df = degree of freedom; Ref = reference; CFI = comparative fit index; RMSEA = root mean square error of approximation; NS = not significant; W1 = 5 days after childbirth; W2 = 1 month after childbirth; W3 = 4 months after childbirth; n = 350 for fathers and 543 for mothers; ** *p* < 0.01, *** *p* < 0.001.

**Table 5 healthcare-11-01740-t005:** Measurement and Structural Invariance of Three MIBS-J Items Across the Three Time Points.

Models		χ^2^	*df*	χ^2^/*df*	Δχ^2^ (*df*)	CFI	ΔCFI	RMSEA	ΔRMSEA	Judgement
Fathers W1 vs. W2 vs. W3	Configural	0	0		Ref	1.000	Ref		Ref	
	Metric	11.250	4	2.813	11.250(4) *	0.987	0.013	0.042	0.042	Reject
	Metric (partial invariance) item 1	9.938	2	4.969	9.938(2) **	0.985	0.002	0.062	0.020	Reject
	Metric (partial invariance) item 6	5.721	2	2.856	5.529(2)	0.993	+0.006	0.042	0.000	Accept
	Scalar	23.686	10	2.369	17.965(8) *	0.975	0.018	0.036	Δ0.006	Accept
	Residual	39.691	16	2.481	16.005(6) *	0.957	0.018	0.038	0.002	Reject
Mothers W1 vs. W2 vs. W3	Configural	0	0		Ref	1.000	Ref		Ref	
	Metric	1.880	4	0.470	1.880(4) NS	1.000	0	0.000	0.000	Accept
	Scalar	55.910	10	5.591	54.03(6) ***	0.950	0.050	0.053	0.053	Reject
	Scalar (partial) item 1	54.097	8	6.762	52.217(2) ***	0.950	0.050	0.060	0.060	Reject
	Scalar (partial) item 6	55.471	8	6.934	53.591(2) ***	0.948	0.052	0.060	0.060	Reject

MIBS-J = Japanese modification of the mother-to-infant bonding scale; *df* = degree of freedom; Ref = reference; CFI = comparative fit index; RMSEA = root mean square error of approximation; NS = not significant; W1 = 5 days after childbirth; W2 = 1 month after childbirth; W3 = 4 months after childbirth; n = 350 for fathers and 543 for mothers. * *p* < 0.05, ** *p* < 0.01, *** *p* < 0.001.

## Data Availability

The data that support the findings of this study are available from the corresponding author upon reasonable request.

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
