# Peer review of "Identifying Core Items of the Japanese Version of the Mother-to-Infant Bonding Scale for Diagnosing Postpartum Bonding Disorder"

_healthcare, 2023, doi:10.3390/healthcare11121740_

Round 1
Reviewer 1 Report
I am pleased to have the possibility to review the study “Identifying Core Items of the Japanese Version of the Mother-to-Infant Bonding Scale for Diagnosing Postpartum Bonding Disorder”. The discussed problem of parent bonding has a meaningful impact on maternal and children’s health. The study population size, the study's character and the statistical evaluation used are undoubtful study strengths.
The introduction section is a well-constructed part of the study. The problem was appropriately introduced to the reader. Nevertheless, the presentation of the scope of the study could be improved, and the aim of the study should be highlighted in the last paragraph of the introduction.
The methodology section was described perfectly and allowed other authors to understand and possibly repeat the performed analysis in other countries.
The result section was presented partly in the discussion section. I recommend moving the figures into the results section and reducing the discussion section to discuss the problem.
Reviewer 2 Report
Thank you for the opportunity to review this manuscript.
This is an interesting topic of study. But there are several elements of confusion that need to be fixed and clarified.
1. Keywords: Please suggest keywords using mesh term search.
2. Abstract: Implications for the Result section need to be more specified.
3. Introduction: It is needed to clarify the limitations of previous studies, including prior research, and why this study should be conducted.
Discussion: The discussion is too short. It is necessary to organize the main contents. Based on the results of this study, it is necessary to revise the discussion by suggesting the significance and direction of the study. It is needed to clarify the limitation of the previous study including prior research including why this study should be conducted. Describe the similarities and differences with the original scale based on previous studies.
Reviewer 3 Report
Comments to the authors
In this manuscript, the authors aim to analyse the measurement invariance of the MIBS-J scale, a Japanese version of the 10-item mother-child bonding scale, adapted to the Japanese population by Yoshida, K. et al. (2012), and structurally analysed by Kitamura, T et al. (2013).
This adapted scale is derived from the MIBS scale, developed by Taylor et al. (2005), introduced from Kumar's (1997) MIBQ questionnaire, initially with 9 items, and subsequently revised by Kumar's colleague Marks, where 3 items were re-worded and a tenth item was added.
In its development, this scale has undergone several changes, both in the number of items and in the re-wording, because of problems of item bias, namely high values of skewness and kurtosis, which have been solved in the above-mentioned publications:
- With logarithmic transformations of the scores to achieve normality, which is necessary in factor analyses when using the ML method.
- When this was not possible, by eliminating the items that did not achieve normality despite the transformation.
- And in the case of eliminating some items, obtaining scale dimensions that did not distribute all the items initially proposed in the scale.
In this manuscript, the authors again use this methodological approach, with which I do not agree, unless they can provide sufficient published evidence to indicate that this procedure is the best solution to the problem of item bias.
In my opinion, this type of methodology leaves the scale validation study unfinished, since when an instrument is developed or adapted, the final result should be used with the totality of the proposed items at the end of the validation. In our case, the scale you are analysing initially had 10 items in its adaptation, and it does not seem correct to me that in the end only 3 items can be used with an invariance of measurement in the three time periods and between genders. These three items are not the initial scale. Without going into the procedure used to analyse the invariance, the three items that "pass" the analysis would be usable in all three times and between the two genders if their validity and reliability properties were analysed again, i.e. the three items would form a new scale revised from the previous one, which should be assessed for content validity, internal reliability (consistency), test-retest reliability, and construct validity, i.e. structural validity, convergent, divergent and/or discriminant validity, and cross-cultural validity.
So, about the methodology used:
- In my opinion, there are no conclusive results that advise eliminating items from a scale when they present high skewness and kurtosis.
- On the other hand, I know of no published results to support a logarithmic transformation over other alternatives.
- For example, Norris, A. E et al. (2004). suggest that data transformation is not always necessary or advisable when calculating Cronbach's alpha or Pearson's product-moment correlation for instruments with asymmetric item responses.
- Can the authors provide any reference where it is advisable to remove items from a scale because of their high skewness, kurtosis, and can they provide any evidence indicating that it is better to use a logarithmic transformation than a correlation other than Pearson's?
I think there are other solutions:
- Revise the re-wording of the items to avoid such important biases.
- Examine the informativeness, difficulty, discrimination of each item using Item Response Theory (IRT).
- Use association coefficients for non-normal data. In particular, if the variables are categorical in nature, such as ordinal ratings, then Spearman's rank-order correlation would be more appropriate than Pearson's r.....
- Other factor extraction methods than ML can also be used: MLR, MLM, MLV ... that are robust to non-normality.
- There is also the possibility with Likert-type rating scales to use polychoric correlations, given the ordinal polytomous nature of the items and their skewness, and to obtain parameter estimates of the CFA models with robust weighted least squares, e.g. WLSMV.
- ...
Finally, the same scale may not be valid for use at all three points in time. When constructing a scale it is important to have clearly defined the construct to be measured, and what would be the conceptual model on which the construct is based and the items to be used for measurement, in this case with a reflective model. In this conceptual model it is important to take into account socio-demographic and temporal characteristics. The construct may be different or change over time, and a different scale may be needed depending on when it is administered.
References
Norris, A. E.; Aroian, Karen J.. To Transform or Not Transform Skewed Data for Psychometric Analysis: That Is the Question. Nursing Research 53(1):p 67-71, January 2004.
Reviewer 4 Report
This paper identified the core items of the Japanese Version of the Mother-to-Infant Bonding Scale for Diagnosing Postpartum Bonding Disorder.
The paper is well structured and analytic, showing the validity of the three-factor model of the questionnaire. I found the paper good and well articulated and I can suggest only minor revisions
Introduction
I suggest adding some quotations and literature discussion on mother-to-infant bonding and the Postpartum Bonding disorder. The topic isn’t fully described, and it should be fully introduced.
Procedure
What about the drop out of mothers in the assessments? There are similar characteristics in the drop-out group that could influence these data?
Method
Probably some more instruments should be adopted to understand the convergent validity with other instrument and more psychometric measures should be assessed such as the reliability of the factors.
Discussion
Add in the limits also the drop out process that could influence the results and the missing of other instruments to validate better the Scale.
Round 2
Reviewer 2 Report
Thank you for the opportunity to review this manuscript. The authors have modified the manuscript well, according to the comments.
Author Response
Thank you very much for your positive comments. Thank you, too, for your cooperation in reviewing our paper.
Reviewer 3 Report
Comments to the authors
First of all, I would like to thank the authors for their effort to respond to my comments.
However, I still think that the logarithmic transformation is not the most advisable option for the high values of skewness and kurtosis they found in the items. The authors mention a paragraph from (West et al., 1995), but there are other issues in this reference that need to be taken into account:
- The chapter aims at non-normal continuous variables, or coarsely categorised variables but not at variables measured on a Likert scale, which are the ones used in the scale analysed. Studies suggest that polychoric correlations should be used when dealing with ordinal data, or in the presence of strong skewness or kurtosis (Muthen & Kaplan, 1985; Gilley & Uhlig, 1993), as is often the case of Likert items.
- There are a number of observations that West et al. (1995) make about the use of transformations (pg. 72), in particular when they say: "transformation of the data changes the original measure y to a new measure y*. The new correlations or covariances are computed between the y* transformed variable, not between the original variables....".
- Moreover, such transformations "...can potentially result in considerable confusion in the interpretation of the transformed results...".
Moreover, subsequent to the work of West et al. (1995), there have been several studies that provide alternatives to the analysis of SEM models when the variables are ordered categorical, using other correlation coefficients and/or other estimation procedures (Norris & Aroian, 2004)(Feng et al., 2013)(Finney & DiStefano, 2013)(Li, 2016).
So, although the discussion seems to me adequate to the results obtained, I wonder whether the conclusions and limitations of the study are appropriate enough to consider this work sufficiently relevant.
In my opinion, perhaps the most important conclusion of this study is the need to revise the items used in this scale to avoid such important biases. As the authors say, this would entail developing a different scale, which can obtain better results in measuring the construct for which the MIBS-J scale was developed.
References
Feng, C., Wang, H., Lu, N., & Tu, X. M. (2013). Log transformation: Application and interpretation in biomedical research. Statistics in Medicine, 32(2), 230–239. https://doi.org/10.1002/sim.5486
Finney, S. J., & DiStefano, C. (2013). Non-normal and categorical data in structural equation modeling. Structural Equation Modeling: A Second Course, 269–314.
Gilley, W. F., & Uhlig, G. E. G. E. (1993). Factor analysis and ordinal data. Education, 114(2), 258–264.
Li, C. H. (2016). Confirmatory factor analysis with ordinal data: Comparing robust maximum likelihood and diagonally weighted least squares. Behavior Research Methods, 48(3), 936–949. https://doi.org/10.3758/S13428-015-0619-7
Muthen, B., & Kaplan, D. (1985). A comparison of some methodologies for the factor analysis of non‐normal Likert variables. British Journal of Mathematical and Statistical Psychology, 38, 171–189. https://doi.org/10.1111/j.2044-8317.1992.tb00975.x
Norris, A. E., & Aroian, K. J. (2004). To transform or not transform skewed data for psychometric analysis: That is the question! Nursing Research, 53(1), 67–71. https://doi.org/10.1097/00006199-200401000-00011
West, S. G., Finch, J. F., & Curran, P. J. (1995). Structural equation models with non- normal variables: Problems and remedies. In T. O. Sage (Ed.), Structural equation modeling : concepts, issues, and applications (p. 289).
